# On a Unified Core Characterization Methodology to Support the Systematic Assessment of Rare Earth Elements and Critical Minerals Bearing Unconventional Carbon Ores and Sedimentary Strata

Scott N. Montross [1,2,*], Davin Bagdonas [3], Thomas Paronish [4,5], Andrew Bean [1,2], Andrew Gordon [1,2], C. Gabriel Creason [1,6,7], Burt Thomas [1], Erin Phillips [3], James Britton [8], Scott Quillian [3] and Kelly Rose [1]

1 National Energy Technology Laboratory, Albany, OR 97321, USA
2 Research Support Contractor, National Energy Technology Laboratory, Albany, OR 97321, USA
3 Center for Economic Geology Research, School of Energy Resources, University of Wyoming, Laramie, WY 82071, USA
4 National Energy Technology Laboratory, Morgantown, WV 26505, USA
5 Research Support Contractor, National Energy Technology Laboratory, Morgantown, WV 26505, USA
6 Oak Ridge Institute for Science and Education, Oak Ridge, TN 37830, USA
7 College of Earth, Ocean, and Atmospheric Sciences, Oregon State University, Corvallis, OR 97331, USA
8 West Virginia Geological and Economic Survey, Morgantown, WV 26508, USA
* Correspondence: scott.montross@netl.doe.gov

**Abstract:** A significant gap exists in our understanding and ability to predict the spatial occurrence and extent of rare earth elements (REE) and certain critical minerals (CM) in sedimentary strata. This is largely due to a lack of existing, systematic, and well-distributed REE and CM samples and analyses in United States sedimentary basins. In addition, the type of sampling and characterization performed to date has generally lacked the resolution and approach required to constrain geologic and geographic heterogeneities typical of subsurface, mineral resources. Here, we describe a robust and systematic method for collecting core scale characterization data that can be applied to studies on the contextual and spatial attributes, the geologic history, and lithostratigraphy of sedimentary basins. The methods were developed using drilled cores from coal bearing sedimentary strata in the Powder River Basin, Wyoming (PRB). The goal of this effort is to create a unified core characterization methodology to guide systematic collection of key data to achieve a foundation of spatially and geologically constrained REEs and CMs. This guidance covers a range of measurement types and methods that are each useful either individually or in combination to support characterization and delineation of REE and CM occurrences. The methods herein, whether used in part or in full, establish a framework to guide consistent acquisition of geological, geochemical, and geospatial datasets that are key to assessing and validating REE and CM occurrences from geologic sources to support future exploration, assessment, and techno-economic related models and analyses.

**Keywords:** rare earth elements; core characterization; critical minerals; coal; sedimentary rock; geochemical analysis; data science

## 1. Introduction

The United States is heavily reliant on imports of certain mineral commodities that are vital to domestic manufacturing, security, and economic prosperity. This dependency coupled with an unstable and insecure source of the commodity creates a strategic vulnerability for supply chains to manufacturing for domestic industry and military needs. A lack of a viable domestic source also limits US competitiveness in several growing markets, particularly clean energy technologies. The importance of these critical materials has been

underscored by several Executive Orders issued by the US Federal Government [1], including the most recent declaration of a national emergency and review of America's critical mineral (CM) and material supply chain [2]. In recent years, the US has produced sufficient policy to benefit the development of domestic sources of REE and CM resources [3].

A critical mineral (CM) is defined as a non-fuel or mineral material that is essential to the economic and national security of the United States, its supply chain is vulnerable to disruption, it serves an essential function in the manufacturing of a product, and its absence would have significant consequences to the domestic economy or national security [1]. The rare earth elements comprising the lanthanide group elements, La-Lu + Y (REE), are also included under the same definition and vulnerabilities as the critical minerals in the order. Conventional sources of REE and CM resources in the US are relatively limited, with some, such as gallium and graphite, being 100% imported from foreign sources [1]. Coal seams around the world where coal is considered a promising source of lanthanides and other critical elements and minerals [4,5] provide a promising potential CM resource chain, if paired with reliable resource evaluations to best understand these materials. The US has significant deposits of sedimentary rocks, including coal and carbon ore deposits, that have been documented in case studies to host REEs [6,7]. In addition, published studies indicate that CM resources occur in mine waste products, such as acid mine drainage, sludge, tailings and other materials currently viewed as waste byproducts from mines [8–11].

At present, there is no systematic method or approach to predict and identify REE and CM resource potential and occurrence from sedimentary systems, carbon ores, or mine waste streams. This lack of a systematic assessment method inhibits our ability to predict where these occurrences are likely to be found and quantify the in-place or economically accessible volumes of unconventional REE and CMs in domestic sedimentary, carbon-ore, and mining waste byproducts. Additionally, lack of a systematic approach to evaluate coal sediment systems for REE and CM has yielded unreliable comparisons between coal regions and studies therein. To develop and establish a domestic supply of unconventional REE and CM, a systematic, data-driven, science-based assessment method and approach is required. Currently, there are limited assessment methods for REE and CM exploration and resource evaluation, this is mainly due to the disparity, or lack of curated and standardized data records for REE and critical minerals in geologic systems. Using a strategic and geologically informed sampling plan of existing coal and sedimentary rock cores from federal and state geological surveys can collect data that are representative of differing geographic, stratigraphic, and coal deposit types.

This work demonstrates the application and utility of a robust and systematic method for collecting core-scale characterization data that can be applied to studies on the contextual and spatial attributes, the geologic history, and lithostratigraphy of sedimentary basins. Our methods reflect the importance of scale and data availability and observations that are crucial to accurately evaluate the specific resource of interest. Furthermore, the characterization data here are crucial to support the development and validation of a proprietary NETL assessment tool for determining the potential for enrichment and occurrence of REE and CM in sedimentary coal systems [12]. New techniques developed over the last three decades have produced high-quality, high-resolution, and often non-destructive data characterizing physical, geochemical, and compositional properties in great detail (for paleoenvironmental data, often to the decadal and sometimes even the sub-annual scale) [13]. Continuous profiles from cores can improve core and heterogeneous rock characterization, improving geologic core descriptions and heterogeneous rock characterization via continuous profiles of core properties. Understanding heterogeneity (scale-dependent) is important for hydrocarbon production and recovery, as well as modeling and extrapolation of observations from one scale to another. Suarez-Rivera et al. [14] utilized a variety of methods and high-resolution measurements—core scale heterogeneous rock analysis (texture and composition), thin section microscopy, strength, thermal conductivity, CT atomic number, and XRF mineralogy—along with multivariate statistical analysis to define rock classes [14]. There are a few studies that define methods for core sampling and core

analysis; however, these methods are not tailored to coal or characterizing carbon ore and concentrations of REE and CM [15–17]. Dai et al. [18] demonstrated that a combination of evidence from geochemistry, mineralogy, palynology, and petrology of coal and from stratigraphy, sedimentology, and sedimentary facies of related rocks is necessary for accurate and comprehensive determination of depositional environments. Glick and Davis [19] stated that careful investigation of individual samples is necessary to verify the statistical results and to follow the basic finding that single sample database entries are not sufficient to fully develop a working statistical geochemistry model that can be further interrogated and used as a tool for prediction or assessment of an individual coal seam, basin, or region. The work by Glick and Davis [19] enforces the requirement for a more systematic and holistic approach at characterization and not simply relying on either indirect proxy or direct geochemical evidence that may not be high enough "n", or simply statistically insignificant.

There is a significant gap in our understanding of the spatial occurrence and extent of REE and CM resources in sedimentary strata, including coal strata. The gap is due to the lack of a cohesive database that is specific to REE and CM that accounts for analytical results from core/rock sample analysis and contextual geospatial distribution of the results, specific to REE and CM. Limitations to the US Geological Survey's COALQUAL database [20]; all US coal basins are not equally represented by the data. Unconstrained sample depths that limit commercial production by not being able to perform selective mining of "promising" resources that will require depth and thickness constraints to target enriched horizons are among some initially recognized gaps. In addition, the COALQUAL database only considers in place coal and not coal refuse disposal piles and fills. The entire PRB extent excluding Native Lands in Montana was recently assessed by the USGS taking advantage of extensive well log information from the coalbed methane boom in the region [21]. With 1.15 trillion short tons remaining in the ground [22], the original in-place coal resource in the PRB is enormous. The PRB also has the highest yearly production of low S coal accounting for approximately 40% of the total US coal production [23]. Prior assessments focused on only a few regional coal beds and the derived net coal thickness maps; however, the recent reserve estimates expanded the number of coals to 47 and incorporated more than 20,000 newly available public-domain drill hole records as well as another 8000 proprietary drill hole records. The additional data enabled the unification of coalbed names and stratigraphy across the PRB [21,24] and the development of a multi-coal bed geological model.

Coal refuse exceeds 10 million tons in Virginia alone, and in the Powder River Basin (PRB) of Wyoming and Montana (the largest coal basin in the US) [21,22], huge volumes of coal waste materials are stripped and backfilled in mines without consideration of geochemical or spatial summary. These data and knowledge are key to both assessing and predicting the occurrence and volumes of REE and CM but also to support future separations and extraction technologies to ensure their commercial viability as a domestic resource. Systematic resource assessment methods are proven to work, and in consideration of ore deposit type, REE and CM occurrences are the only way to verify total resource estimate. For example, from the 1800s to the 1970s, the oil and gas drilling success rate was 50% or less. After development of the petroleum system method in the 1980s, the exploration success rate increased to above 90% at present [25].

As described above, DOE's unconventional REE and CM resource assessment efforts are addressing technology and innovation gaps to help expedite establishment of a domestic REE and CM research and technology development program that supports the identification of new, unconventional REE and CM resources using data-science enriched methods and tools developed by NETL to simultaneously establish a robust domestic supply while beneficiating legacy byproducts stemming from historical carbon-ore and mining activities. The work is chief to informing the future, improved techno-economic estimates of domestic, unconventional REE and CM supplies and strategies to support research, commercial, and policy decision making. Ultimately, leveraging strategic partnerships and collaborations with key stakeholders (e.g., federal and state geologic surveys,

industry, and academic partners) can expedite development, testing, technology transfer and commercialization of the unconventional REE and CM assessment method.

## 2. Materials and Methods

Core descriptions, subsampling, and analysis of material was performed in a systematic manner using a workflow that can be broken into four sub-tasks (Figure 1 and Table 1). The sub-tasks include:

1. Core collection and field descriptions;
2. Core logging and data collection;
3. Discrete subsampling and microanalysis;
4. Data analysis, visualization, and interpretation.

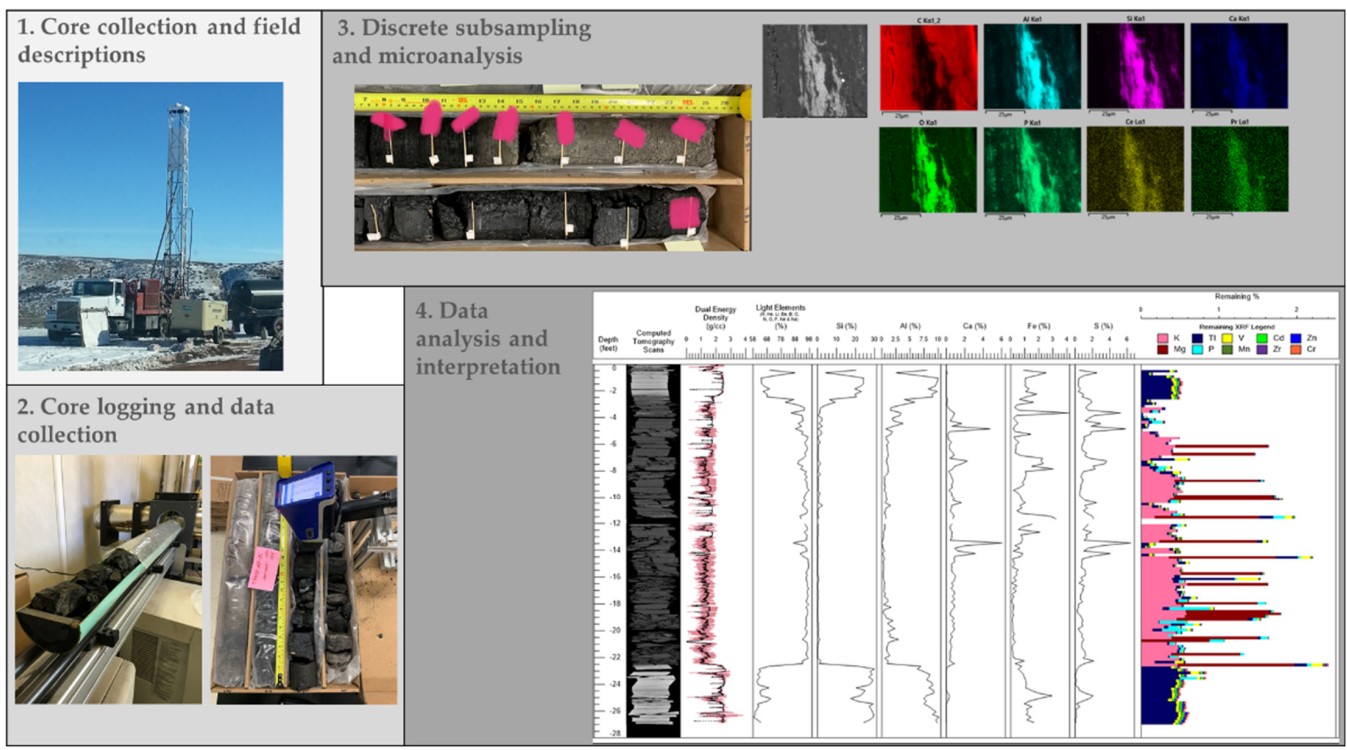

**Figure 1.** Core characterization workflow steps and methods for collection and analysis of data from sedimentary rock cores. 1. Core collection at Brook Mine, Sheridan Wyoming. 2. MSIL core logging and pXRF analysis of cores in the NETL laboratory. 3. SEM imaging and EDS mapping of elements. 4. Compiled core log detailing the Mining-plus suite elemental results; (from left to right) CT images, dual energy density, mean density (black), standard error (red) (gm/cc), light elements (up to silicon) (LE) (%), silicon (Si) (%), aluminum (Al) (%), iron (Fe) (%), calcium (Ca) (%), sulfur (S) (%), remaining elements (%).

Attribute and data collected are designed for input into a NETL-developed assessment model for REE and critical minerals in sedimentary systems [12]. Data tables for organization and submission of data can be found at NETL EDX Data Worksheet.

NETL and CGER core characterization workflow, data collection and formatting are designed to produce a machine learning-ready dataset that can be integrated into the NETL-developed assessment model for REE [12] or other data-driven assessment tool. The complete data input table and descriptions of data required and level of importance, defined as critical, important, or if known, are shown in the supplemental material section and can also be accessed on the world wide web at NETL EDX Data Worksheet. A summary of the steps and methods to acquire the data and information needed for geological samples from a drill core or geolocated hand samples used in NETL's URC Assessment Method [12] are shown in Figure 1 and Table 1. The spreadsheet comprises the relevant metadata needed

to ensure proper context for the sample data used in an assessment of REEs in coal and other sedimentary systems, as well as fields for the data inputs.

**Table 1.** Core characterization, data acquisition, and attribute categories for systematic collection and data analysis of geomaterials for studies on the occurrence of rare earth elements and critical minerals in sedimentary strata.

| Characterization Step | Data Acquisition Method | Attribute Category * |
|---|---|---|
| Core collection, field and laboratory processing | Collate information from drillers log about: location, core quality, core length and depth of collection, observed litho or stratigraphic markers, and brief description of macro features | Sample ID, sample location, site characteristics, collection information, other metadata |
| Core logging and data collection | Photo documentation, detailed undisturbed core description, CT scanning, multi-sensor core logging, pXRF logging | Chemical analysis |
| Discrete subsampling | Systematic subsampling of core based on data collected during core descriptions and logging | Other metadata |
| Microstructure and microanalysis | Thin section petrography, SEM/EPMA imaging and advanced image analysis; elemental analysis by SEM/EPMA-EDS or WDS, solid-phase geochemical analysis (ICP-OES/MS, TGA, XRD, XRF) | Chemical analysis |
| Data analysis, interpretation, and curation | Production of digitized lithostratigraphic and geochemical logs, digitization of data into files formats for geospatial, statistical analysis and machine learning ready formats | Site characteristics, chemical analysis |

* Attribute category from NETL-EDX Sample Collection Datasheet.

### 2.1. Core Collection, Field, and Laboratory Processing

Cores were drilled and collected from two locations in the Wyoming PRB (see Figure 1). Full core sections or individual subsamples were obtained by NETL-RIC researchers through partnerships with Ramaco Carbon LLC and the University of Wyoming School of Energy Research, Center for Economic Geology Research (CEGR). The cores were collected specifically for characterization and analysis of REE along a vertical section of the coal seams. Material collected and subsampled included coal and non-coal sedimentary rock both above and below the coal seam of interest.

### 2.2. Core Logging and Data Collection

Core logging and data collection followed methods developed by NETL and CEGR (Supplemental Material Section S1). Briefly, the intact cores were measured, viewed, and photographed. Noteworthy observations and lithologies present were documented along with discrete subsample locations for geochemical and microanalysis. Core logging data collected by CEGR included photographs, visual observations, stratigraphic and lithologic logs, drill logs, and observations of primary and secondary minerals visible in hand samples.

NETL advanced core logging data included vertical (through the core depth-wise) CT scans, orthogonal CT splices, and XRF data (major and trace elements). Core scale CT scanning was performed with a medical Toshiba® Aquilion TSX-101A/R medical Scanner, Tustin, CA, USA. The compiled core logs were scaled to fit on single pages for rapid review of the combined data from the medical computed tomography (CT) scans and X-ray fluorescence (XRF) measurements. Dual energy CT scanning was used to approximate the density [26,27]. A portable handheld Innov-X® X-Ray Fluorescence Spectrometer, Waltham,

MA, USA, was used to measure relative elemental abundances. The Mining-plus suite, Waltham, MA, USA, was run at a 6 cm (~0.2 ft) resolution at 60 s per beam exposure time over for the entire core length. The Mining-plus suite utilized a 2-beam analysis that resolved major elements (Mg, Al, Si, P, S, Cl, Fe, K, Ca, and Ti), minor elements (V, Cu, Ni, Cr, Mn, and Pb), trace elements (Co, Zn, As, Zr, Mo, Ag, Cd, Sn, Sb, Hf, W, and Bi), and an aggregated "light element" (H to Na). A second set of XRF data was collected from core and hand samples using a handheld Olympus Vanta M-series portable X-ray Fluorescence Spectrometer, Tokyo, Japan, with 3-beam capability. The analysis was performed on core samples that were air dried for 24 h. The scan surface was prepared by removing the outer 2-3 mm of the sample surface with a file or scraper to provide a fresh, flat surface for analysis. Scans were conducted on the cleaned outer edge of the core and on a 1" thick face from the interior. Each scan was run for 10 or 20 s per beam exposure (e.g., Beam 1, 10 kV; Beam 2; 20 kV; and Beam 3, 50 kV). The 3-beam analysis resolved the following elements (Mg, Al, Si, P, S, Cl, Fe, K, Ca, and Ti), minor elements (Sr, V, Cu, Zn, Ni, Cr, Mn, and Pb), trace elements (Ba, Ce, La, Nd, Y, Co, Zn, As, Zr, Mo, Nb, Ag, Cd, Sn, Sb, Hf, Th, U, W, and Bi), and an aggregated "light element" (H to Na). General detection limits for REE were La 50ppm, Ce 65ppm, Y 1 ppm, and Nd 80 ppm on a whole rock basis. The whole rock concentration (in ppm) of La, Ce, and Y were used for individual sample screening and core logging analysis to identify REE enrichment zones and inform subsampling for ICP-MS analysis. These non-destructive analyses were completed prior to the cores being sub-sampled for various geochemical and microanalytical analyses.

*2.3. Discrete Subsampling and Microanalysis*

Two methods—ore body and selective sampling—were used to collect subsamples along the entire length of the core for geochemical and microanalytical analyses. Both core subsampling methods are adopted from traditional core characterization efforts that are enhanced with an ore body assessment strategy. Cores were collected and analyzed at CEGR using the ore body sampling method. A complete coal core sampling procedure developed by CEGR is shown in Figure 2. On the collected core, sample collection grids are initially laid out across the entire core length prior to sampling as would be standard for "ore-body" mapping in a vertical distribution (vertical core). By bracketing the coal seam(s) of interest, complete vertical grids can be constructed through the entire "ore-body" of interest, as well as providing clues to depositional history and potential modes of REE- and CM-enrichments in discrete zones within the coal measure or adjacent sedimentary rock over- and under-burden.

The second method, the selective sampling strategy, utilizes sample grids and a selection of subsampling points from the core that are based on changes in lithologic and geochemical properties. Subsample locations are guided by the identification of horizons in the core with elevated REE concentrations (La, Ce, Nd, and Y) by rapid analysis of the material with a portable XRF unit. Discrete sub-samples are collected from different depths along the entire length of the core based on the initial lithology and geochemical survey. For larger seams (>20 cm) subsamples were collected at 5–10 cm intervals throughout the coal depending on the thickness of the seam, at higher resolution (1 cm)-approaching boundaries, and at depths that exhibit major textural and/or geochemical changes in the material. The ore body method is described for adoption to the DOE-funded CORE-CM effort that utilizes the method for high-resolution data acquisition of coal sediment systems. Both methods are appropriate for data collection necessary for submission of core characterization data to the NETL-EDX REE-CM Workspace and for use in the NETL Unconventional Rare Earth and Critical Minerals (URC) Assessment model (see Table 1).

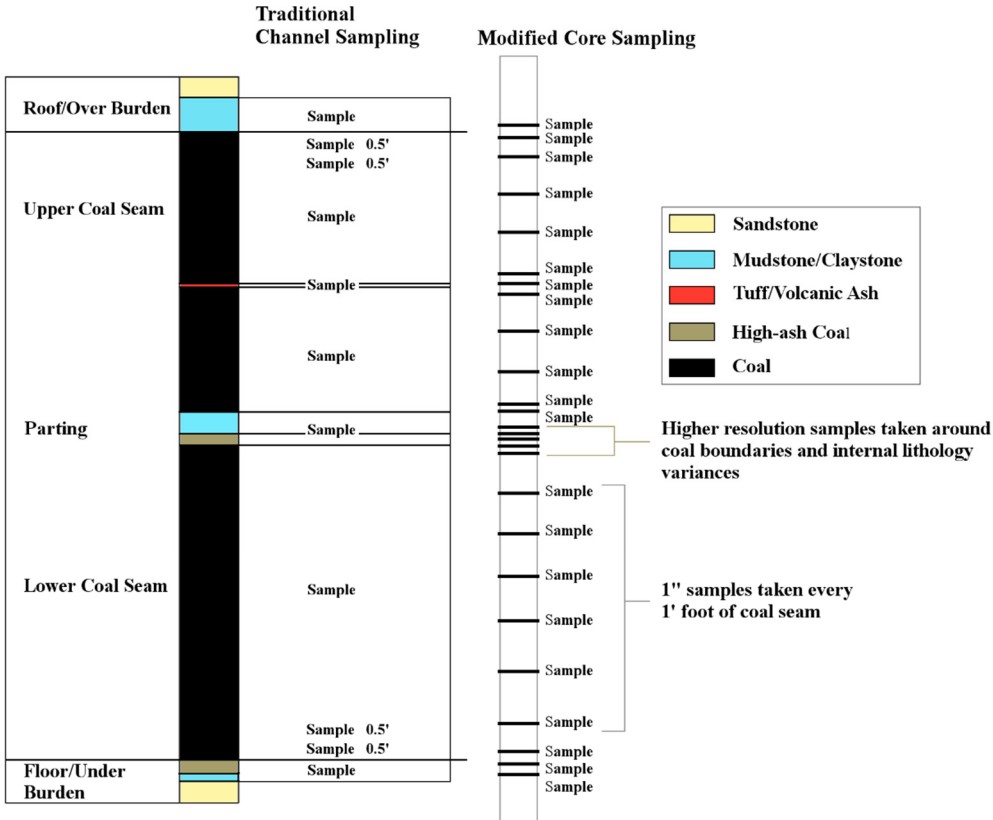

**Figure 2.** Example of sample spacing in coal core with sediment variances and expected REE and CM anomalies around those variances. Comparison is given to tradition channel sampling of the same stratigraphic sequence. Modified from CEGR sample plans provided by Coalgeo. LLC.

Core subsample splits collected by CEGR and NETL researchers were shipped to NETL (Albany, OR) for processing. Sample bags were catalogued in a spreadsheet to cross-reference the core log and to confirm that all samples were present and labeled with the correct depth. In the laboratory, small (approximately 15–20 g) subsample splits were crushed to a powder with a ceramic mortar and pestle and reserved for determination of major elements and mineralogy and geochemical analysis at the NETL Pittsburgh Analytical Laboratory (PAL). The remaining material was archived. Solids analysis consisted of determination of moisture and ash content by thermogravimetric analysis (TGA) and major, minor, trace, and REE concentrations by inductively coupled plasma-optical emission spectrometry (ICP-OES) or mass spectrometry (ICP-MS) follow methods outlined in Bank et al. [28]. All trace elements including REE were quantified using ICP-MS.

Petrographic thin sections were made from the remaining core sample. The thin sections were viewed and imaged with a polarized light microscope and a scanning electron microscope (SEM). Backscattered electron (BSE) images were collected from single fields of view and multiple fields combined into a large area image. SEM imaging and EDS analysis was performed following methods in Thompson et al. [29] for the analysis of REE in coal and coal byproducts. Microscopy and microanalysis data were used to assist with putative identification of inorganic minerals present in the coal and rock layers. Point spectra, line spectra, and elemental maps were collected and used for displaying the distribution of elements, determining the composition of individual mineral grains and coal, and for aiding in the identification of mineral phases present in the sample. Large area BSE images (2 mm × 2 mm) were collected from representative areas of the thin sections and were used to compare textural properties and quartz grain sizes between sample depths.

*2.4. Data Analysis and Interpretation*

All characterization and core logging data were digitized and visualized using commercially available geologic data software programs, Strater version 8.7.19 by Golden Software ( and WellCad version 5.4 by Rockware (Golden, CO, USA). The software allows for the creation of digital lithostratigraphic and geochemical depth logs. The logs are used for correlation and visualizing the distribution of REE and identifiable mineral phases across different lithologies or rock textures encountered in the core. All data logs organized by depth were created in MS Excel and imported into Strater or WellCAD.

## 3. Results

Vertical depth profiles of lithologic properties were combined with geochemical data to create chemostratigraphic logs of each core analyzed. The digital logs were created using commercially available software from RockWare Inc. (Golden, CO, USA). Results from geochemical analyses, including microscopy and microanalysis, were used in conjunction to determine the predominant mineral phases in the subsamples collected. The suite of characterization data was used to develop paragenetic sequence(s) for mineral deposition based on spatial and textural relationships observed in the material. The paragenetic sequences provide information on the plausible mechanisms that lead to REE or CM enrichment within the coal measure or adjacent sedimentary strata.

Correlative chemostratigraphic and lithological log plots show the variations in lithology with changes in geochemical properties by depth (see example in Figure 1). XRF and ICP-MS results are plotted as depth logs, with key elements displayed in individual tracts (left to right: light elements, Si, Al, Fe, Ca, S, and REE). The next ten most abundant elements are represented in the remaining elements tract. The tract displays the remaining percent from 0 to 2.5% with the area under the curve separated by the proportion of each individual element.

Results from geochemical analyses were used to calculate key geochemical ratios and indictors. Spider plots and element cross plots were used to identify relationships between REE occurrence and concentration with other key geologic parameters. Examples of geochemical indicators or proxies used to evaluate the occurrence and potential mechanism(s) for REE enrichment in geologic material include comparison of REE concentrations (total or individual elements) to total ash (proxy for amount of inorganic material in the coal) and HREE/LREE ratios (example shown in Figure 3), or specific REE element anomalies (Figures 4 and 5).

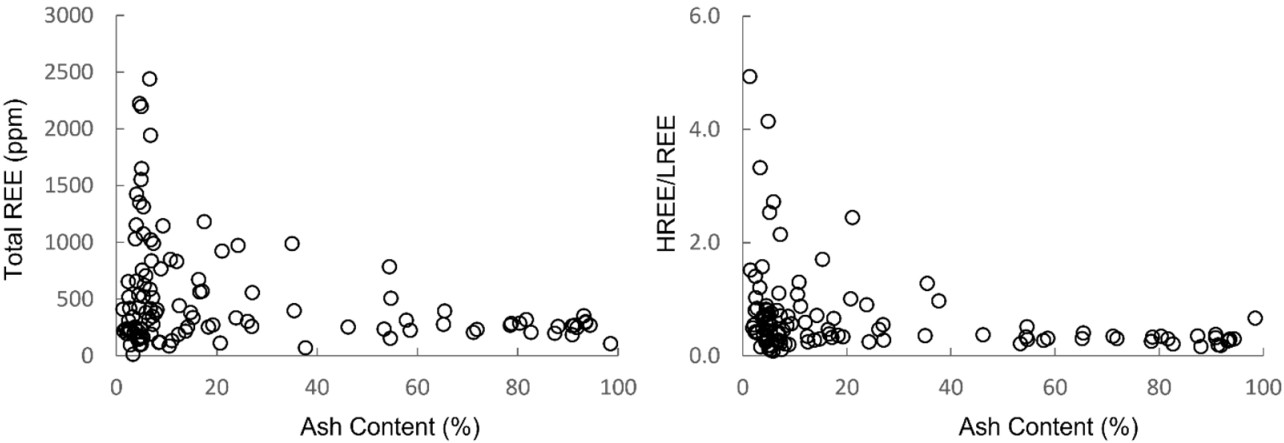

**Figure 3.** Cross plots (**left**) total REE in ppm on ash basis and (**right**) HREE/LREE ratios versus ash content.

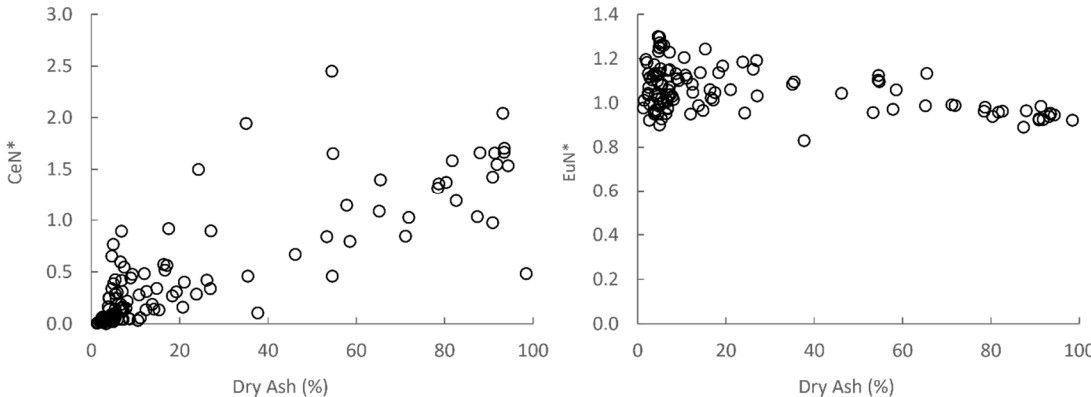

**Figure 4.** Plots of REE anomalies Ce (**left**) and Eu (**right**) from core samples. Ce$_N$ is the ratio of the concentration of Ce in the sample to Ce in the upper continental crust (UCC). Ce$_N$* and Eu$_N$* are calculated using the equation(s) from Dai and others, 2016 [5] where Ce$_N$* = 0.5La$_N$ + 0.5Pr$_N$ and Eu$_N$* = 0.5Sm$_N$ + 0.5Gd$_N$.

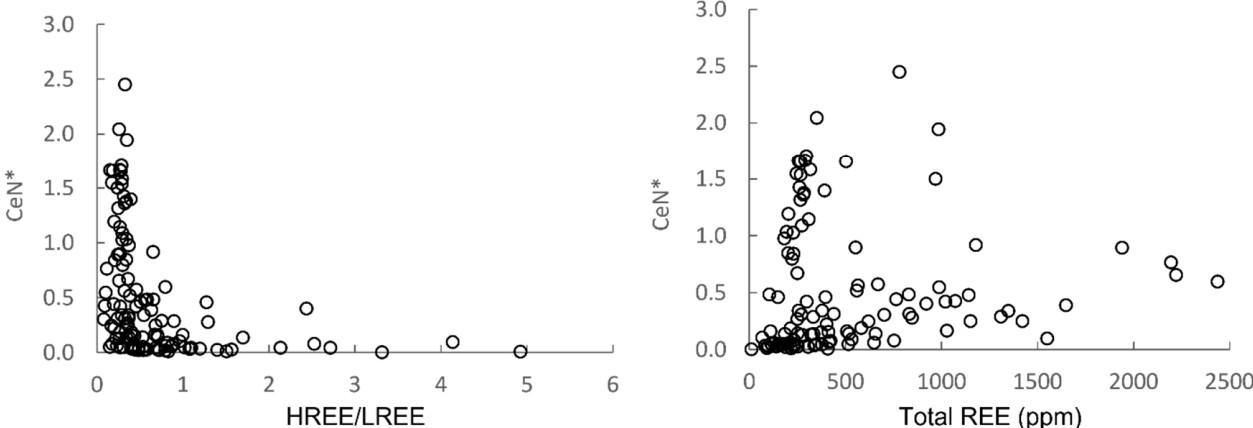

**Figure 5.** Plots of Ce$_N$* versus HREE/LREE ratio (**left**) and total REE in ppm (**right**).

Geochemical REE anomalies in coal are attributed to sediment source, hydrothermal water, volcanic ash inputs [4,5,30], and both primary and secondary mineral species susceptibility of hosting available cations over geologic time periods. Low-resolution sampling may not provide the resolution to identify various REE anomalies in order to determine the source of REE in the coal or sedimentary rock.

Weathered tuff ash inputs, a key input of REE to the PRB coals in this study, would be expected to be a basin-wide occurrence (e.g., [31]). Other datasets such as COALQUAL may become useful for extending the range of the enrichment-associated anomaly, depending on the availability of REE data. In the case of Ce$_N$* or Eu$_N$*, as shown here, his quantitative analysis of La, Ce, Pr, Nd, Eu, Sm, and Ho by ICP. (Figures 4 and 5). Not only are these calculations important for determining the source or enrichment of certain REE, but they are also used to determine the quality of the analytical data [5]. Core samples analyzed here show a range of Ce$_N$* values when compared to the concentration of mineral material in the coal (e.g., as ash content) or REE concentration. A europium anomaly (as Eu$_N$*) was not present in the samples analyzed (Figure 4).

Core depths with a cerium anomaly present had the lowest ratio of HREE/LREE of the subsamples analyzed. Core depths with a Ce-anomaly and low (<0.4) HREE/LREE may represent zones where CeIII is oxidized to CeIV due to interactions with oxygenated groundwater, resulting in the mobilization and displacement of the remaining REE.

The coupling of geochemical and microstructural data from SEM imaging and EDS analysis aids in mineral phase identification and to make basic determinations of diagenetic

and paragenetic relationships. For example, the BSE image and accompanying EDS map from a coal sample containing high REE (>2000 ppm on an ash basis) is shown in Figure 1. The material in the coal is concentrated in sub-millimeter pore spaces or fractures within the coal. The fine-grained mineral material is composed of quartz, other felsic grains (Al, Si-rich) with varying amounts of trace elements and a variety of clay minerals. Calcium, P, Ce, La, and Y are co-located within the Al-Si matrix. The felsic grains are sub-micron and are presumably weathered volcanic ash based on the elemental composition. Grains of REE-P such as monazite and xenotime, most 1–5 microns in length, were observed as individual crystals in the samples and typically occurred in or around the felsic/clay mineral layers. The predominant Fe mineral phases are pyrite framboids and iron oxide grains. Pyrite framboids and iron oxide were observed together in pore spaces and between mineral layers at the top and bottom of the coal seams near the contact with non-coal bearing clay or other lithology present. Partial digestion textures and replacements Fe-S (pyrite) and Fe-S-O (iron oxide) of pyrite were observed in fractures and pore spaces, conversion to a fracture and/or pore filling cement was noted. The digestion textures and conversion of Fe-S to Fe-S-O is indicative of pyrite weathering via sulfide oxidation. This is likely due to the increase in oxygen fugacity in the system plausibly due to the introduction of meteoric water into the coals. In the presence of oxygen, iron sulfur species are converted to ferric iron oxyhydroxide, or another similar oxidized Fe solid phase, whereas the pyrite in the original coal formed under anoxic conditions present during coalification.

## 4. Discussion

Efforts to identify the most promising REE and CM resources began with a broad effort to sample and analyze materials from a variety of sources across the coal and mining value chain, inclusive of raw geologic samples. However, in 2017, the program also recognized that without a reliable domestic source and supply of unconventional REEs and CMs, such as those from carbon ores and their byproducts, these separations technologies would remain reliant on the global markets and sources. Thus, in 2018, NETL initiated development of the URC REE-CM assessment tool to systematically predict and assess domestic deposits of REEs from carbon ore and other sedimentary systems (see [7,12]). This method is the first of its kind, using a big-data, machine-learning-enabled geoscience approach to improve prediction and identification of high-concentration deposits of REE and CM in sedimentary, carbon-ore based systems. Similar to resource assessment methods and approaches that have previously been developed to improve successful discovery of economic deposits of other types of mineral resources (e.g., oil, gas, gold, copper), this method is designed to improve discovery and quantification of unconventional REE and CM deposits. The methodology relies on three phases: the first phase is data collection and editing, as described here. The second phase includes modeling, coalbed correlation, and identification of areal restrictions. The final phase represents the coal resource and reserve analysis with assignment of reliability, including measured, indicated, inferred, and hypothetical zones. Zones can be lithostratigraphic zones or enrichment zones that are identified by elemental composition and quantity.

NETL and its partners will continue to focus on constraining which unconventional REE/CM are most relevant to establishing a domestic supply and most prospective from domestic coal deposits. While the resource assessment efforts target identifying in-place, technically and potentially economically recoverable scales of evaluation at the mine, seam and basin scale, they rely upon finer-scale data and studies as well. Core, outcrop and mine scale data and information are key to constraining and understanding how and why unconventional REE and CM form in these types of systems and the processes and conditions that result in the vertical and lateral heterogeneity of these deposits. In addition, contextual geologic data and information at these finer scales are important for constraining the lithologic, structural, and secondary alteration history of these systems [32] to improve predictions of the occurrence and concentrations of unconventional REE and CM deposits at all scales.

At present, much of these data and knowledge either do not exist in forms appropriate for supporting unconventional REE/CM assessments or are incomplete. For example, the COALQUAL database is often cited as relevant to meeting these needs, but it was collected for a different original purpose, and the homogenized nature of the analyses significantly limits the utility of that data for these types of assessments. The timing of much of the collection and analysis means that the data are not of the quality or quantity expected now. Other assessment studies and programs such as the USGS methods for data collection and assessment of coal beds utilize seam-based geometries using core data. COALQUAL is the largest public database containing coal-quality information. The USGS released the first version of COALQUAL in 1994 and the second in 1998 [33,34]. With the byproducts and mine waste systems, it is also key to understand the human engineering and history of these systems. With sufficient data and knowledge from these types of studies, the predictability and accuracy of the Unconventional REE and CM assessment method will be greatly improved. While these methods provide volume of coal or carbon ore in a particular basin or smaller sub-basin, this type of analysis does not yield reliable numbers for the assessment of REE or CM within the seams. Single bulk measurements of coal seams are valuable for tracking bulk trends in basins across the US; however, these techniques are not at enough resolution, nor do they provide accurate assessments of areas of enrichment of REE or CM, as observed in the case studies shown here. In addition to the described data sample and sourcing, including databasing of the existing COALQUAL record, approximately 12,000 additional coal samples have been analyzed from the 1990s to the present, and the USGS is in the process of verifying the data for eventual release. This addition may yield useful reference data, but again, sample strategy for COALQUAL database inclusion has not been tailored for REE and CM ore body assessment purposes.

At present, the lack of geologically referenced and targeted samples means there is not sufficient data to improve predictions on the distribution and controls on REE concentrations in sedimentary systems, specifically coal-bearing strata. Geospatial analysis, including the use of geostatistics, has increased the success for discovery of promising geologically hosted resources, such as water, oil, and gas. The occurrence and volume of these resources are not random, they are a product of geologic processes over time. Thus, characterizing and determining occurrence and extent of REE in sedimentary strata in a strategic and geologically referenced approach expedites improvements to prediction and evaluation of commercial volumes of REE resource and aids in future prospecting and assessments.

A strategy to evaluate coal core selectively across the PRB in order to overcome the described limitations of COALQUAL and mine-reported coal data began in 2018 [35,36]. The initial coal core review included complete and partial core samples and associated data provided by the USGS to CGER (USGS Core Research Center, Denver, CO (CRC). The samples and accompanying data included reliable geolocation (x,y,z) and sufficient material for geochemical analysis including trace elements and REE. Importantly, this coal core included non-coal portions of the sediment system. However, USGS-hosted materials proved to be limited for studies on the basinal properties of the ore body and considerations; thus, direct coal core collection from host mine sites has become the most useful tool for filling data gaps and broadening resource determinations, including testing predictability of estimate occurrences.

Direct coal core collection from target zones and core that is randomly available from various mines have provided relevant assessment opportunities, including REE and CM methods development for coal bearing strata and assessment in both selectable and random locations throughout the PRB. Direct on-site retrieval of coal-bearing core provides complete intervals of coal strata including the bounding sediments both above and below the target coal seams. Additionally, target coal seams have been those already produced for thermal coal use, thus representing the actual resource potentially available from current mining operations/locations, as resource assessment results become relevant for this consideration.

Coal-bearing cores from both random and selected locations were initially assessed by grid sampling, as would be standard for ore body extent determination. This grid was initially set at single-foot intervals including several feet above and below the coal seams. Initial geochemical results including REE and CM were then evaluated, and refining of grid sample spacing (inch-scale grid) was conducted in both the REE and CM relatively enriched and relatively depleted core segments. Specifically, REE-concentration anomalies appeared consistent with preliminary investigations (e.g., [35]), despite large variances in coal seam identity and location.

Select coal core locations were devised to benefit perceived gaps and verify the apparently predictable REE behaviors described in the initial investigation. These select cores were drilled parallel to each other with less than 155 m of spacing, within a well-constrained coal mine host, and bearing the same coal sediment system boundaries. Both grid sampling, as described above, and selective sampling around predicted REE enrichments were conducted to test relevance of grid sampling in mine-scale coring operations. As expected, with preliminary background knowledge of REE and CM anomalies in familiar/previously sampled PRB coal seams, the grid-sampled core showed elevated REE and CM concentrations in both the bounding anomalies and internal anomalies [35]. The selective sampling of the immediately adjacent core produced REE- and CM-elevated concentrations with highest total values but missed resolving the entire anomaly range in the described bounding and internal anomaly locations. Grid sampling continues to provide the most relevant "ore body" extent descriptions, while not surprisingly, selective sampling can sometimes produce the larger REE and CM concentration value for an "ore body" location [36].

The enrichment zones identified in Wyoming PRB coals were identified through the systematic characterization of the multiple cores that led to the identification of vertical zones of enrichment of REE. The vertical zones are also traceable between cores and highlight the horizontal extent of the resource. The data are inclusive and are at a high enough resolution that we can trace geochemical evidence that supports mechanisms for the enrichment of REE in the coals.

In addition, the correlation of well and core data across the basin was completed using linear and circular cross sections in StratiFact. PC/cores were used to produce gridded coal bed models and avoid false zeroes due to well data too short to intersect. Coalbed grids output to ASCII/ArcView. Technical restrictions to mining include mined-out areas, burned coal areas, land-use restrictions, technical restrictions, overburden to-coal-bed ratios, stripping ratios, resource reliability categories, and coal bed depth. Previous economic assessments applied a standard vertical projection downward through the coal beds when applying regulatory surface buffers (such as a 300 ft buffer around an inhabited house) below the surface. Because of the additional setback distance required to maintain a safe mining pit highwall angle, a restricted area widens with depth when surface-mining operations are considered.

Technical issues were encountered while processing large datasets with multiple formats and units. For example, a small adjustment of resolution was needed prior to and during the lithologic and stratigraphic column construction. Depths were not consistent between the different records of data. The XRF core logging data were listed in hundredths of feet (e.g., 37.25 feet) and split into increments of 0.2 feet, whereas the original core drilling logs were listed in tenths of feet and rounded to the nearest half a foot (e.g., 37.5). A combination of two conventions was used for visual observations and lithologic descriptions, depths were listed in tenths of feet and rounded to the nearest half a foot, sample locations were also in tenths of feet but were in inconsistent increments. While this is seemingly a trivial discrepancy, when inputting the data into Strater or WellCAD, it led to offsets between the lithostratigraphic log, the CT scans, sample locations, and XRF data. There are also some small depth discrepancies between the orthogonal CT image slice images that were used in Strater and the images used to denote sample locations. This is noted in the figures in the footnotes.

The coupled efforts of characterization of geologic samples and creation and validation of the URC REE-CM tool, a novel big-data, machine-learning-enabled geoscience approach, improve prediction and identification of promising unconventional deposits of REE and CM in sedimentary, carbon-ore based systems. This NETL-led effort has benefitted from key collaborations with industry, university, USGS and state surveys. These external partnerships have been key to knowledge and data generation from strategic analysis of samples at the local scale, to integration of commercial data and resources at the core (tens of meters), mine (100's of meters), and basin (kilometers) scale to drive assessment and discovery. Resource assessments are key to the success of the separation technologies and system economic assessment aspects of DOE's portfolio, providing key insights and data about unconventional REE and CM source volumes, host materials, ore minerology, and locations that can be paired with appropriate separations and extraction technologies to drive commercial success and information vital to constraining domestic techno-economic assessments for policy and commercial decision makers.

## 5. Conclusions

The methodology supports the development and curation of big-data driven, geospatial modeling to help beneficiate and optimize REE and CM potential from primary (carbon-ore) and secondary (byproducts and waste streams) sources. This strategic, geologically informed sampling and characterization effort will accelerate DOE's goals of maturing the efficient and data-driven, knowledge-driven assessment and prediction of REEs in sedimentary systems in support of developing a future economic domestic REE resource. This workflow provides a focused, finer-scale look at the distribution and occurrence of REE and CM across a vertical profile of coal- and non-coal-bearing rock. The core characterization workflow discussed here provides the following benefits to REE and CM assessment and exploration.

1. The methodology is fully integrated with REE and CM data submission into the NETL Energy Data Exchange (EDX), a publicly available dataset developed to support development and validation of the NETL URC assessment tool for key US sedimentary basins. The effort supports the development and curation of big-data driven, geospatial modeling data in a machine learning format to help beneficiate and optimize REE and CM potential from primary (carbon-ore) and secondary (byproducts and waste streams) sources.
2. The selective sampling method for collecting and analyzing subsamples from lithological contacts and coal zones with increased mineralization provides sufficient data to generate and test hypotheses on the mechanism for REE enrichment. The strategic coring efforts and lithogeochemical and stratigraphic correlations are employed to increase spatial (x.y) coverage for lateral seam mapping. Data collected at this resolution are used to ground truth e-geophysical logs at mine scale.
3. Coupling of core level descriptions with geochemical analyses, including SEM and x-ray microanalysis, permits the identification of mineral phases (primary and secondary), identification of volcanic ash layers, quantification of bulk REE and composition of REE-bearing mineral phases, and provided evidence of diagenesis and paragenetic relationships with pyrite dissolution coupled to the weathering of volcanic ash and reformation of iron-sulfur oxide mineralization.
4. Our results and observations indicate that more than one geologic control or mechanism exists in REE and CM enrichment and deposits within a carbon-ore and surrounding geologic media [35].

**Supplementary Materials:** The following supporting information can be downloaded at: https://www.mdpi.com/article/10.3390/min12091159/s1. Section S1: Systematic core analysis method for coal and non-coal coal bearing sedimentary lithotypes; Section S2, Table S1: Prioritization of data and information collected for geologic samples.

**Author Contributions:** Conceptualization, S.N.M., K.R., C.G.C., B.T. and D.B.; methodology, S.N.M., K.R., C.G.C., T.P., A.G. and D.B.; formal analysis, S.N.M., A.B. and K.R.; resources, K.R. and S.Q.; writing—original draft preparation, S.N.M. and D.B.; writing—review and editing, S.N.M., D.B., E.P., J.B., C.G.C., T.P., A.G. and A.B.; supervision, K.R., B.T., J.B. and S.Q.; project administration, K.R., B.T. and S.Q.; funding acquisition, K.R. and S.Q. All authors have read and agreed to the published version of the manuscript.

**Funding:** The work was performed in support of the National Energy Technology Laboratory's ongoing research under the Rare Earth Element Field Work Proposal DE-FE-1022420 by NETL's Research and Innovation Center, including work performed by Leidos Research Support Team staff under the RSS contract 89243318CFE000003.

**Data Availability Statement:** Data supporting the reported results can be found at https: www.netl.edx.gov (accessed on 1 April 2021).

**Acknowledgments:** The partnership between NETL and Ramaco Carbon is through a CRADA cooperative research agreement. We acknowledge Nicole Rocco for processing core samples in the laboratory, Paige Morkner for assistance with SEM imaging of core samples, Randy and Charlie Atkins and the staff at Ramaco Carbon and Western Water Engineering in Sheridan Wyoming for their continued support in providing core and developing the method, and Dustin Crandall (USDOE-NETL) and members of the NETL Morgantown CT scanning laboratory for CT analysis and multisensory core logging efforts. Additional samples and funding have been provided by the University of Wyoming School of Energy Resources to support both NETL and CEGR efforts.

**Conflicts of Interest:** The authors declare no conflict of interest.

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
