# Peer review of "On a Unified Core Characterization Methodology to Support the Systematic Assessment of Rare Earth Elements and Critical Minerals Bearing Unconventional Carbon Ores and Sedimentary Strata"

_minerals, doi:10.3390/min12091159_

Round 1

Reviewer 1 Report

This manuscript is apparently the result of studying only one core, but many techniques are used for it. However, the manuscript does not hold much attention and does not seem to be able to provide a universal study protocol or method for such studies, contrary to its title! Many of the methods used in it are those that are used in the classical methods of studying cores. In my opinion, a very important weakness of the proposed method is that it requires equipment and techniques, some of which are not easily available everywhere, but are also very expensive! From the point of writing view, the manuscript is not very well written and there is a lot of excessive material in it (especially in the introduction and discussion sections) which is boring and annoying! Overall, I do not recommend accepting it.

1. As it is said in the review report, the authors tried to provide a comprehensive method to study coresin prospecting and pre-evaluating for rare earth elements and specific minerals.Their proposed procedure is not attractive, in my opinion, and does not answer any important question regarding the description of boreholes.In addition, the techniques used are usually expensive and not
available everywhere.

2. The topic seems original and relevant in the field, but it does not address a specific gap in the field.

3. I think it does not add anything to the subject area compared with other published material.

4. Authors should use method(s) that are readily available and inexpensive.These methods should lead us quickly and directly to the answer to the desired question (are there any rare earth elements or specific minerals in the core at hand?), while this is not the case.

5. The conclusions are consistent with the evidence and arguments presented, but they do not address the main question posed, in my opinion.

Author Response

Review 1
1. As it is said in the review report, the authors tried to provide a comprehensive method to study coresin prospecting and pre-evaluating for rare earth elements and specific minerals.Their proposed procedure is not attractive, in my opinion, and does not answer any important question regarding the description of boreholes.In addition, the techniques used are usually expensive and not
available everywhere.
Response: A table of all data entries is now included in the Supplemental information. There are a number of techniques shown/described in the manuscript, this should be taken as the best case scenario, the characterization here is extensive, however, the critical data required in order to develop a dataset for modeling or machine learning use is actually quite simple. Pleae refer to the supplemental table that shows the different levels of importance for data (critical, important, if known). The descriptions, logging of features, basic stratigraphic markers can be done with little to no cost. Standard chemical analysis either by xrf and ICP-MS is routine and is necessary in order to determine the chemical composition.

2. The topic seems original and relevant in the field, but it does not address a specific gap in the field.
Response: There is a significant gap in the availability of REE and critical mineral data that contains geospatial/geolocational data that is well defined and can be used in initial investigations into resource evaluation at mine or larger basin scale. There are existing datasets housed by USGS that contain a wealth of coal quality data (COALQUAL) however in many cases critical elements (REE) are not captured. Our method provides two ways to capture this data, pXRF onsite at core collection or strategic subsampling and analysis using XRF (screening)  or ICP-MS for full quantification.  

3. I think it does not add anything to the subject area compared with other published material.
Response: To our knowledge, there are no other core characterization workflows that relate to studies on presence of REE or critical minerals in the literature. This includes studies on unconventional systems. There are similar studies published on oil and gas cores for reservoir characterization, however, the data collected and scope are different wrt evaluation of sedimentary rock for REE/CM. The methods shown here are already being employed by multiple industry/exploration company partners in the PRB and CentApp systems, as well as DOE CORE-CM program and university partners. 

4. Authors should use method(s) that are readily available and inexpensive.These methods should lead us quickly and directly to the answer to the desired question (are there any rare earth elements or specific minerals in the core at hand?), while this is not the case.
Response:  The methods shown to collect "critical" data (see list provided in the Supp material are inexpensive and only require basic core description skill.

5. The conclusions are consistent with the evidence and arguments presented, but they do not address the main question posed, in my opinion.
Response: The main question posed is how and what data is necessary to collect in order to evaluate the potential occurrence of REE in sedimentary systems. We believe the work presented here and the conclusions demonstrate this

Reviewer 2 Report

Reviewer comments:

Please find my comments on the marked Word copy of the manuscript.

Author Response

Review 2

The authors revised the current manuscript based on the reviews/comments provided in the attached document peer-review-21522326.v2.pdf. Grammatical edits and wording changes to the revised manuscript can be found in the track changes on the draft.

Reviewer 3 Report

Review of, Scott N. Montross et al :

On a unified core characterization methodology to support the systematic assessment of rare earth element and critical mineral bearing unconventional carbon ores and sedimentary strata

This is an interesting paper to read and from my point of view it is well written and easy to read.

The core of the paper is chief to informing future, improved techno-economic estimates of domestic, unconventional REE and CM supplies and strategies to support research, commercial, and policy decision making.

Below I give some additional comments listed with chapter name.

Abstract

No comments

Introduction

Need to say, the above technologies in this study are not omnipotent “magic wands” that can be used alone to support the systematic assessment of rare earth element and critical mineral bearing sedimentary strata

Materials and Methods

Other new technologies  could help also in understanding and ability to predict occurrence of unconventional REE/CM resources; such as XRF core scanner, Hyperspectral imaging, Micro-XRF, …..etc.

The X-ray fluorescence (XRF) core scanner provides the most rapid, non-destructive high-resolution elemental measurements than portable handheld Innov-X® X-Ray Fluorescence Spectrometer, that used in this study. A new generation of XRF core scanners provide High Resolution Geochemical data set and  the spectral data in continuous motion along the entire length of the drill core. Indeed, REE element or CM profiles along the cores could help in understanding the geochemical element content and distribution

 Results

The results section is where you report the findings of your study based upon all  methodologies you applied to gather information. But in your result chapter, you concentrated only in the results from geochemical analyses with no explanation for other methods used in the study. Such as X-Ray Computed Tomography Scanning data , drill core logs, Microstructure and microanalysis, Visualization and structural measurements.

Discussion

The study lacked clear control groups or comparison between the methods that used in this study as between XCT-XRF and lithogeochemistry. In the discussion section, authors need to explain the comparison between qualitative and quantitative research methods and highlights differences and similarities

Conclusions

Obviously, this chapter has to be re-written when the comments above are addressed.

References

The references list should follow referencing style of the journal.

Some references should be included in this study

 Luth, S., Sahlström, F., Bergqvist, M., Hansson, A., Lynch, E.P., Sädbom, S., Jonsson, E., Andersson, S.S. and Arvanitidis, N., 2022. Combined X-Ray Computed Tomography and X-Ray Fluorescence Drill Core Scanning for 3-D Rock and Ore Characterization: Implications for the Lovisa Stratiform Zn-Pb Deposit and Its Structural Setting, Bergslagen, Sweden. Economic Geology.

Luth, S., Sahlström, F., Jansson, N., Jönberger, J., Sädbom, S., Landström, E., Bergqvist, M., Arvanitidis, N. and Arvidsson, R., 2019. Building 3D geomodels using XRF-XRT-generated drillcore data: The Lovisa-Håkansboda base metal-and Stråssa-Blanka iron deposits in Bergslagen, Sweden. In 15th SGA Biennial Meeting 2019, Glasgow, Scotland.

Warlo, M., Bark, G., Wanhainen, C., Butcher, A.R., Forsberg, F., Lycksam, H. and Kuva, J., 2021. Multi-scale X-ray computed tomography analysis to aid automated mineralogy in ore geology research. Frontiers in Earth Science, 9, p.1222.

Butcher, A.R., 2020, July. Upscaling of 2D mineralogical information to 3D volumes for geoscience applications using a multi-scale, multi-modal and multi-dimensional approach. In IOP Conference Series: Materials Science and Engineering (Vol. 891, No. 1, p. 012006). IOP Publishing.

 General recommendation

The paper needs a minor revision before acceptance.

Good Luck

Review-v1

Author Response

Grammatical changes were made and are shown on MS Word track changes. The reviewer provided a pdf with comments and edits that were incorporated into the revised manuscript. The comments (grammatical and reference to figures) are shown in the track changes. 
Additional response:
Figure 1 was revised to show high resolution image of data tracks and core photos.
The supplemental material includes step by step instructions for core preparation and descriptions.

Round 2

Reviewer 1 Report

The manuscript is not changed except for some writing corrections and improvement of its English quality. So my general opinion is what I stated in the review of the previous version.